Developing bottom drifters to better understand the stranding locations of cold-stunned sea turtles in Cape Cod Bay, Massachusetts

Page Felicia M. 1 felicia_woods@uri.edu
Manning James 2
http://orcid.org/0000-0002-6050-7351 Howard Lesley 1
Healey Ryan 1
Karraker Nancy E. 1
1 Department of Natural Resources Science, University of Rhode Island , Kingston, Rhode Island , United States
2 Northeast Fisheries Science Center, National Oceanic and Atmospheric Administration , Woods Hole, Massachusetts , United States
Florinsky Igor
Electronic publication date: 2023 Aug 30
Publication date: 2023
Volume: 11
Electronic Location ID: e15866
Received 2022 Jun 30; Accepted 2023 Jul 17
Copyright: © 2023 Page et al.
Copyright year: 2023
Copyright holder: Page et al.
License: This is an open access article distributed under the terms of the Creative Commons Attribution License, which permits unrestricted use, distribution, reproduction and adaptation in any medium and for any purpose provided that it is properly attributed. For attribution, the original author(s), title, publication source (PeerJ) and either DOI or URL of the article must be cited.
License URL: https://creativecommons.org/licenses/by/4.0/

Keywords: Drifters, Sea turtles, Cold stunning, Cape Cod Bay, Stranding

Funding: National Science Foundation Graduate Research Fellowship Program Sophie Danforth Conservation Biology Fund University of Rhode Island Enhancement of Graduate Research Award This material is based upon work conducted by a researcher supported by the National Science Foundation Graduate Research Fellowship Program. Funding for supplies was provided by the Sophie Danforth Conservation Biology Fund, awarded by Roger Williams Park Zoo, and the University of Rhode Island Enhancement of Graduate Research Award. The funders had no role in study design, data collection and analysis, decision to publish, or preparation of the manuscript.

==============================
Every fall, juvenile sea turtles in the Northwest Atlantic Ocean are threatened by rapidly declining water temperatures. When sea turtles become hypothermic, or cold-stunned, they lose mobility—either at the surface, subsurface, or the bottom of the water column—and eventually strand at the shoreline where rescue teams associated with the Sea Turtle Stranding and Salvage Network may search for them. Understanding the effects of ocean currents on the potential stranding locations of cold-stunned sea turtles is essential to better understand stranding hotspots and increase the probability of successful discovery and recovery of turtles before they die in the cold temperatures. Traditional oceanographic drifters—instruments used to track currents—have been used to examine relationships between current and stranding locations in Cape Cod Bay, but these drifters are not representative of sea turtle morphology and do not assess how bottom currents affect stranding locations. To address these knowledge gaps, we designed new drifters that represent the shape and dimensions of sea turtles—one that can float at the surface and one that sinks to the bottom—to track both surface and bottom currents in Cape Cod Bay. We found a marked difference between the trajectories of our new drifter models and those that were previously used for similar research. These findings bring us one step closer to identifying the transport pathways for cold-stunned sea turtles and optimizing cold-stunned sea turtle search and rescue efforts in Cape Cod.

Introduction

The ecological significance of sea turtles extends well beyond their roles as predator and prey and their contributions to the health of the world’s oceans (Bouchard & Bjorndal, 2000), yet six of the seven extant species are considered vulnerable, threatened, or critically endangered (IUCN, 2020). Since 1978, extensive conservation efforts have been underway to bring Kemp’s ridley sea turtles (Lepidochelys kempii)—the world’s smallest and most endangered sea turtle species—back from the brink of extinction (Shaver, 2005; Caillouet, Shaver & Landry, 2015; Shaver & Caillouet, 2015; Wibbels & Bevan, 2019). Bi-national and multi-agency collaborative programs such as the Kemp’s Ridley Sea Turtle Restoration and Enhancement Program and the Sea Turtle Stranding and Salvage Network (STSSN) have brought communities together to rescue and protect sea turtles for over 40 years. After a decline of over 99% in nest production from historical records (1947–1985), the efforts of these conservation programs have resulted in an increase from 702 nests recorded in 1985 to nearly 25,000 recorded in 2017 (National Research Council, 1990; Spotila, 2004; Shaver et al., 2005; Bevan et al., 2016; Wibbels & Bevan, 2019). Although these endeavors have shown promising results, Kemp’s ridley sea turtles remain critically endangered (Wibbels & Bevan, 2019).

Since Kemp’s ridley sea turtles have the most restricted distribution of all sea turtles and have historically nested almost entirely in the Gulf of Mexico (for exceptions see Johnson et al., 1999; National Park Service, 2018), conservation-related research has primarily focused on addressing threats contributing to declines in adults and nests—e.g., equipping fishing vessels with turtle excluder devices, protecting nests from poachers and predators, and translocating eggs (National Research Council, 1990; Shaver, 2005). Juvenile sea turtles have received little attention in previous decades but are currently a focus for sea turtle conservation in the northeastern United States. The nutrient-rich waters of the Northwest Atlantic Ocean serve as an important foraging ground for juvenile Kemp’s ridley sea turtles (Lazell, 1980; Morreale & Standora, 2005), where thousands of individuals may congregate in the Gulf of Maine to feed during warmer months (Spotila, 2004). The region is also notorious for unpredictable weather—such as nor’easters and frequent cold snaps—during the late summer and fall months. As a result, these juvenile turtles engage in a risky tradeoff between optimizing foraging during a crucial developmental phase and the threat of hypothermia if they delay migration to southern overwintering habitats (Spotila, 2004; Morreale & Standora, 2005).

The biggest threat to juvenile sea turtles in the Gulf of Maine and its southernmost embayment—Cape Cod Bay, Massachusetts—is severe hypothermia, commonly referred to as cold stunning. Cold stunning occurs when water temperatures drop below roughly 10 °C and cause physiological processes to begin shutting down (Still, Griffin & Prescott, 2005; Shaver et al., 2017; Liu et al., 2019). Once cold-stunned, sea turtles are unable to actively swim and may die from prolonged exposure to the cold temperatures, whether in the water or on the beach, or by drowning because they cannot raise their heads out of the water (Shaver et al., 2017). It is believed, following a sudden cold snap, that some proportion of turtles become incapacitated and remain buoyant at the surface either because of a lack of ability to dive or because gases build up from undigested food in the gut (B Still, 2018, personal communication). Other sea turtles either dive below the surface—where the water temperature is more stable—and remain there or lose their ability to swim and sink to the bottom. Observations of injuries and shell conditions showed that many turtles drag along the bottom before washing up. Mortality rate among cold-stunned Kemp’s ridley sea turtles in Cape Cod Bay is approximately 40–50% and largely affects turtles 0.75–4 years old, with straight-line carapace lengths of 20–30 cm (Avens et al., 2020). Although cold stunning is not a threat unique to temperate waters (e.g., Witherington & Ehrhart, 1989; Shaver et al., 2017), it impacts hundreds of threatened and endangered sea turtles in Cape Cod Bay every fall—including Kemp’s ridleys, loggerheads (Caretta caretta), and green turtles (Chelonia mydas)—of which Kemp’s ridley sea turtles comprise the majority of those recovered (see Supplemental Information).

For several decades, the STSSN has collaborated with the Wellfleet Bay Wildlife Sanctuary of the Massachusetts Audubon Society in the U.S. and has trained volunteers to patrol Cape Cod Bay beaches by foot in search of stranded sea turtles. Cold-stunned sea turtles are carried toward the beaches by winds and currents, where they are typically found by these search teams shortly after high tide when the water is receding. However, the Cape Cod beaches where sea turtles strand extends over 100 km, which requires that volunteers search large areas to find cold-stunned sea turtles as quickly as possible to reduce exposure time. Over 1,000 stranded turtles were recovered from Cape Cod beaches in 2014 and 2020, and stranding numbers are expected to increase with a changing climate (Griffin et al., 2019; Moise, 2021). It is also important to note that population recovery—estimated to be rebounding by as much as 19% per year because of conservation efforts (Spotila, 2004)—will increase the number of sea turtles exposed to threats (Putman, Hawkins & Gallaway, 2020) and coincidentally increase the number of sea turtles that strand from cold-stun events. Reducing the amount of time that cold-stunned sea turtles are exposed to potentially lethal air temperatures is crucial to recovery, and the ability to predict where sea turtles are likely to strand in each storm event or cold snap may help focus search efforts and increase the likelihood of survival.

Previous research on cold stunning in the Northwest Atlantic Ocean examined the importance of environmental correlates, such as temperature and wind direction, as spatial and temporal drivers of sea turtle cold-stunning and stranding locations (Burke, Standora & Morreale, 1991; Morreale et al., 1992; Still, Griffin & Prescott, 2005; Liu et al., 2019). Other studies have estimated circulation patterns in Cape Cod Bay based on sediment transport from Massachusetts Bay (Beşiktepe, Lermusiaux & Robinson, 2003; Warner, Butman & Dalyander, 2008) and particle tracking models (Liu et al., 2019), but information is limited on the effects of these currents on sea turtles themselves. With the exception of research by Liu et al. (2019), wind direction has been the primary variable used to estimate the locations of sea turtle strandings in Cape Cod Bay.

Wind is a principal driver of water currents at or near the ocean’s surface (i.e., surface currents) and is often used to estimate the trajectory of objects floating in the water (Garrison, 2013). However, other factors contribute to the flow of water, especially in shallow water embayments like Cape Cod Bay. For example, the effects of waves and tidal oscillation are not captured when wind direction is the sole driver used to model drifting objects. Ocean currents are often studied using drifters—oceanographic instruments used to track ocean currents via satellite telemetry—to analyze these trajectories over time (Novelli et al., 2017) and offer a more accurate representation of ocean circulation patterns.

To simulate ocean currents in Cape Cod Bay, Liu et al. (2019) compared data from moorings, sea turtle stranding locations, and satellite-tracked ocean surface drifters to validate a model that investigated the cause and transport of cold-stunned turtles. Their study addressed questions regarding the impact of wind-driven surface currents on potential sea turtle stranding hotspots but the effect of currents on cold-stunned sea turtles that have sunk to deeper waters is still largely speculative. It is unknown whether the buoyancy of cold-stunned sea turtles changes once they are immobilized—they may float at the surface of the water (positively buoyant), below the surface (neutrally buoyant), or sink to the bottom (negatively buoyant).

Previous research has modeled potential stranding hotspots by examining the influences of wind-driven surface currents on drifters that float on or just below the surface, but poorly represented the size and shape of the sea turtles that typically cold stun (see drifter dimensions in Table 1 below). Research that has incorporated the use of satellite-tagged sea turtle carcasses, in addition to wooden effigy drifters, has shed light on the seasonal variability in stranding patterns (Reneker, Cook & Nero, 2018; Cook et al., 2021), has developed backtracking models to estimate the location of at-sea mortality (Nero et al., 2013), and assessed how decomposition rate impacts sea turtle drift (Reneker, Cook & Nero, 2018; Nero et al., 2022). Although these studies provided a useful foundation, we have little understanding of how other environmental factors influence stranding patterns, particularly for turtles that have sunk below the surface. The objectives of our study were to (1) design new drifter models that are morphologically representative of sea turtles, (2) examine the effects of surface and bottom currents in Cape Cod Bay on the transport of these drifters, and (3) compare drifter stranding hotspots to sea turtle stranding hotspots during the cold-stunned sea turtle stranding season. This research may help focus search and rescue teams on beaches with higher stranding potential under cold stunning conditions, reduce the exposure time for stranded turtles, and ultimately improve the chances of rescue and recovery of cold-stunned sea turtles.

Table 1 Dimensions and mean drift times for drifters deployed during the 2019 stranding season for cold-stunned sea turtles.

Drifter design	Height
(m)	Length
(m)	Drift depth*
(m)	Number of drifters
(n)	Drift time (h)	
Mean	Range	SD	
Mini-boat	1.36	1.52	0.91	5	5	3−6	±1.30	
Davis-style	1.88	1.22	−1.55	6	174	58−325	±112.90	
Sea turtle surface	0.13	0.36	0.00	4	16	8−24	±8.16	
Sea turtle bottom	0.13	0.36	−11.5	4	160	44−371	±150.34	
Note:

* Drift depth refers to the deepest point the drifter reaches. For the mini-boat, this is the height of the sail rather than the portion that is submerged below the water.

Methods

To quantify differences between surface and bottom currents and determine how those currents influence stranding locations, we documented trajectories and endpoints of four types of drifter models in Cape Cod Bay, Massachusetts. No animals were involved in the sampling, so no special permissions were required for this research.

Study site

Cape Cod Bay is a semi-enclosed embayment surrounded by the hook-shaped peninsula of Cape Cod, Massachusetts. The bay is approximately 1,564 km2 and reaches a maximum depth of 62.8 m. Currents in the bay tend to flow counterclockwise but are driven largely by wind patterns and vary by season. Although the waters of the bay are stratified in the summer, they are well mixed from late fall through the winter months because of strong seasonal winds (Signell & List, 1997).

Drifter designs

We designed a set of drifters to serve as more realistic models of sea turtles and deployed them simultaneously with traditional drifters to target currents at different depths throughout the bay. A deployment group consisted of a standard Davis-style drifter (Davis, 1985), sea turtle-shaped surface drifters (three were used in case one or more did not collect consistent data), a drogued sea turtle-shaped bottom drifter, and an unmanned miniature sailboat. Each drifter was outfitted with a satellite transmitter (Globalstar SmartOne Asset Tracker) or a GPS data logger (Canmore GT-730FL-S SiRF IV edition) that allowed us to record the drifter’s path.

Davis-Style Surface Drifter—An aluminum-framed adaptation of the Davis-style drifter (hereafter “Davis drifter”) is a standard model used in oceanographic research to track ocean currents. Like the Coastal Ocean Dynamics Experiment “CODE” drifter (see Poulain, 1999; Liu et al., 2019), the body of the Davis drifter consisted of an aluminum central mast, four spars, and four canvas cloth sails, in addition to an acorn buoy and platform to hold the satellite transmitter—programmed with a one-hour sampling rate—above the water (Fig. 1). This design, with the aluminum frame, was selected because of the low cost to refurbish and reuse on subsequent deployments.

Figure 1 Davis-style surface drifter used to track currents in Cape Cod Bay, Massachusetts.

(A) Drifter before deployment to show size comparison. (B) Deployed surface drifter shows main body submerged. Photo credit: Chip Carroll (A) and Felicia Page (B).

Sea Turtle Surface Drifter—The sea turtle surface drifters (hereafter “surface drifter”) were designed to mimic juvenile Kemp’s ridley sea turtles in size (20–30 cm straight-line carapace length), shape, and weight (3–5 kg). Similar to those used by Santos et al. (2018), the drifter bodies were built from plywood and polystyrene foam board, with a hole cut in the center to add ballast before deploying, and GPS data loggers—with a one-minute sampling rate—were housed in small plastic bottles attached to the drifter bodies (Fig. 2). Because cold-stunned sea turtles are not completely rigid, the flippers were designed using a hinge to attempt to mimic potential movement and resulting drag but, when observed during trials, the flippers remained relatively immobile. Just enough ballast was added to partially sink the drifters below the surface while maintaining positive buoyancy (Fig. 2C).

Figure 2 Sea turtle surface drifter with 25 cm straight-line carapace length used to model sea turtle stranding locations in Cape Cod Bay, Massachusetts.

(A) Bottom of the drifter with ballast compartment. (B) Decorated carapace of drifter with bottle attached for GPS logger housing. (C) Beached sea turtle surface drifters. Photo credit: Felicia Page.

Sea Turtle Bottom Drifter—To form the sea turtle bottom drifter (hereafter “bottom drifter”), we made a plaster mold using the carcass of a cold-stunned Kemp ridley sea turtle that had died (approved under U.S. Fish and Wildlife Service Recovery Permit #1150C-1). The plaster mold was used to prepare a secondary silicone mold before creating the final cast of the body, which consisted of lightweight polyurethane casting resin safe for marine use (Fig. 3). The drifter had a hollowed “belly” to add ballast at the release location to compensate for changes in daily salinity, using only enough weight to create negative buoyancy (4–6 kg total). A retractable tether—adapted from an outdoor retractable PVC clothesline—was used to anchor the bottom drifter to the buoy-mounted satellite tracker—programmed with a 1-h sampling rate (Fig. 3E). The retractable tether helped keep the floating transmitter as close as possible to the submerged drifter while floating through shallower waters.

Figure 3 Making the sea turtle subsurface drifters used to model sea turtle stranding locations in Cape Cod Bay, Massachusetts.

(A) Plaster mold of deceased cold-stunned Kemp’s ridley sea turtle. (B) Silicone casts of sea turtle. (C) Polyurethane resin in mold. (D) Assembled subsurface sea turtle drifter. (E) Deploying the tethered subsurface drifter. Photo credit: Felicia Page.

Unmanned Miniature Sailboat—An unmanned miniature sailboat (hereafter “mini-boat”, Fig. 4), equipped with a satellite transmitter programmed with a one-hour sampling rate, was provided by Educational Passages (Kennebunk, Maine, USA) and the Gulf of Maine Lobster Foundation (Kennebunk, Maine, USA) and was used to track the flow directly above the surface of the water. This device was instrumental in providing a more accurate estimate of the wind conditions nearest to the water’s surface during the drifter deployments and helped guide search efforts for recovering the GPS-equipped surface drifters once they stranded. Since location data were not being transmitted to the satellites for the GPS-equipped drifters, we estimated the landing sites based on the relationship between wind direction and mini-boat landing.

Figure 4 Unmanned miniature sailboat documented wind conditions in Cape Cod Bay, Massachusetts.

(A) Size comparison just before deployment of mini-boat. (B) Mini-boat after deployment. (C) Mini-boat after stranding. Photo credit: Felicia Page.

Observed drifter data

Six drifter deployments, each including a set of all four drifter types, were conducted throughout Cape Cod Bay, Massachusetts between 31 October and 26 November 2019. Drifter deployments took place ahead of storm fronts when temperatures were expected to drop below the cold stunning threshold (10 °C; Spotila, O’Connor & Paladino, 1997; Milton & Lutz, 2003) for sea turtles and winds were expected to exceed 5 m/s (sustained). Drifters were deployed from the eastern side of Cape Cod Bay (near 41.8999°N, 70.1202°W) where the bay was approximately 11 m deep at mean low tide. This location was selected because the depth did not exceed the length of the retractable tether attached to the bottom drifter, allowing it to reach the bottom of the bay. We provided contact information on all drifters and mini-boats in the event that stranded equipment was encountered by beach walkers.

Data collection—Data for the satellite-tracked drifters (Davis drifters and bottom drifters) and mini-boat were maintained and accessed through the ORBCOMM telecommunications network. Since the surface drifters were equipped with GPS data loggers, rather than satellite transmitters, data tracks were downloaded once the units were recovered from the beaches. Satellite information was used to direct the drifter recovery teams to the satellite transmitter-equipped drifters and the mini-boat, and GPS-equipped drifters were primarily recovered by beach walkers and STSSN volunteers while searching for cold-stunned sea turtles.

We observed the data remotely via ORBCOMM for the bottom drifters regularly to determine if the drifter had detached from the buoy, or if the drifter became entangled. Following the guidance of Haza et al. (2018), we observed drift patterns in the satellite data looking for spans of missing data points and changes in drift velocity. Missing data points indicated that the buoy may have flipped over, submerging the satellite transmitter, and detached buoys or entangled drifters responded to wind forcing differently than properly functioning drifters (i.e., detached floating drifters moved faster and entangled drifters showed less movement).

Hourly data for environmental correlates were retrieved from the National Oceanic and Atmospheric Administration’s National Data Buoy Center and a weather station at Provincetown Municipal Airport (Provincetown, Massachusetts, USA). These data were used to estimate the mean wind speed around the time, and immediately after, the drifters were deployed.

Data Analysis—The drifter speed was determined using the distance traveled over the time interval between location points—i.e., Δdistance/Δtime = m/s. Location data were also used to calculate the compass direction for the direction of travel (Table 2). We focused the analysis on the trajectories on the first 4 h after deployment, limited by the average time it took for the mini-boat to strand.

Table 2 The initial direction of travel and speed of drifters and the corresponding wind variables.

Date	Drifter	Compass
direction	Direction (degrees)	Estimated speed (m/s)	
11/14/2019	WIND	NNE	18	4.41	
11/14/2019	Mini-boat	N	4	0.83	
11/14/2019	Sea turtle surface	NWN	326	0.11	
11/14/2019	Davis-style	NW	312	0.11	
11/14/2019	Sea turtle bottom	NWW	306	0.07	
11/19/2019	WIND	ENE	66	2.62	
11/19/2019	Mini-boat	E	87	0.37	
11/19/2019	Sea turtle surface	ES	104	0.15	
11/19/2019	Davis-style	SEE	119	0.14	
11/19/2019	Sea turtle bottom	ESE	114	0.15	
11/26/2019	WIND	NNE	27	8.94	
11/26/2019	Mini-boat	NEN	37	1.01	
11/26/2019	Sea turtle surface	NE	44	0.20	
11/26/2019	Davis-style	NEE	35	0.22	
11/26/2019	Sea turtle bottom	NE	47	0.18	
Note:

The estimated speed is the average of the first four hourly readings for the period after the drifters were deployed.

Comparing hotspots of drifter strandings to sea turtle strandings

The STSSN collects data each winter on cold-stunned sea turtles, including location and condition (dead/alive), as rescue teams recover stranded turtles. Data for the 2019 Cape Cod Bay sea turtle stranding season were provided by the Massachusetts Audubon Society. STSSN volunteers recovered a total of 299 cold-stunned sea turtles, dead and alive, from the beaches of Cape Cod Bay during the 2019 stranding season—from 9 November 2019 to 3 April 2020.

Locations of high-density stranding locations (hotspots) were identified using a kernel density analysis in ArcGIS, for both the stranded sea turtles and the drifters. Drifter stranding data were grouped by deployment date and drifter type, and the deployment dates were compared to the 2019 sea turtle stranding data. Since drift time varied by deployment date, sea turtle stranding data were restricted to either the latest date the drifters stranded or by 1 week—whichever was more restrictive.

Results

Drifter designs

Except for one of the Davis drifters—which was swept out of the bay and lost at sea—nearly all of the Davis, surface, and bottom drifters and the miniboat were recovered after stranding. GPS-equipped sea turtle surface drifters were recovered by beach walkers over an 8-month period but some were not found because they were not satellite-tracked. Drift time differed between drifter types. We documented that, on average, drift time was 10 times longer for Davis drifters than for surface drifters (Table 1). Similarly, drift time was 10 times longer for bottom drifters than surface drifters (Table 1).

Observed drifter data

Three of the six deployment clusters produced sufficient data for comparison—at least one of each drifter model transmitted consistently from these three clusters—because either the transmitter did not communicate with the satellite or the GPS data logger was damaged and the data could not be retrieved. For example, one of the bottom drifters was slightly too heavy and pulled the transmitter below the surface, preventing communication with the satellites, and rough seas also contributed to loss of communication at times. The waterproof housing on some of the GPS data loggers also failed, causing the USB connector to rust from the saltwater, making the data irretrievable. The missing data for these three deployments prevented comparisons between the surface and bottom drifters.

Although the effects of currents varied by wind conditions (Table 2), there were marked differences in the trajectories of the traditionally used Davis drifters, surface drifters, and bottom drifters. The sea turtle surface and bottom drifters moved in distinctly different patterns throughout the duration of drift from deployment to stranding. Despite a greater difference in depth between the two sea turtle-shaped drifters (surface and bottom), we observed more separation between the Davis drifter and the sea turtle surface drifter than between the two different sea turtle drifter models (Fig. 5). The degree of divergence between the tracks varied under different wind conditions, but, regardless of date of deployment, the data exhibited noticeable differences in the trajectories of the four drifter models. Hotspots for the strandings of the bottom drifters were south of the hotspots of the surface drifters.

Figure 5 Drifter tracks following the 19 November 2019 deployment in Cape Cod Bay, Massachusetts.

With the exception of the mini-boat, which stranded after 4 h, the drifter tracks shown here are limited to the first 24 h after the deployment to provide a simplified visualization of the degree of separation between the different drifter models.

Comparing hotspots of drifter strandings and sea turtle strandings

Several drifter sets were deployed during the week with peak stranding numbers associated with cold stunning in 2019. The stranding hotspot for all drifter models (Fig. 6A) was centered in Truro, Massachusetts, northeast of our deployment site. A majority of the 299 sea turtles stranded during the winter of 2019 were recovered in Barnstable, Massachusetts (n = 69, 23%) and other hotspots (Fig. 6B). The stranding hotspot for the surface drifters (Fig. 7A) was also in Truro, ~12 km north of the bottom drifter hotspot (Fig. 7B) in Wellfleet, Massachusetts.

Figure 6 Stranding hotspots for the sea turtle-shaped drifters and sea turtles in Cape Cod Bay, Massachusetts, 2019.

Hotspots highlight areas where the largest numbers of (A) drifters or (B) cold-stunned sea turtles were recovered throughout the season. Red indicates the highest number of data points, while yellow indicates intermediate and blue indicates the lowest number.

Figure 7 Stranding hotspots for sea turtle-shaped drifters in Cape Cod Bay, Massachusetts, 2019.

(A) Sea turtle surface drifters (n = 14) stranding hotspots. (B) Sea turtle subsurface drifters (n = 4) stranding hotspots. Red indicates the highest number of data points, while yellow indicates intermediate and blue indicates the lowest number.

When comparing the sea turtle-shaped drifter strandings to the cold-stunned sea turtle strandings for the season (Fig. 8) we saw an overlap in stranding locations but not necessarily the hotspots. For example, the stranding locations for the drifters deployed on 14 November were centered in the outer Cape (Fig. 8A), while the sea turtle strandings for the week of 14 November were centered in the mid-Cape (Fig. 8B). Of the different drifter models, the bottom drifter stranding hotspots were closest to the 2019 stranding hotspot for cold-stunned sea turtles.

Figure 8 Comparison of sea turtle-shaped drifter and cold-stunned sea turtle stranding hotspots in Cape Cod Bay, Massachusetts from 14–18 November, 2019.

(A) Drifters (n = 6) deployed on 14 November. (B) Cold-stunned sea turtle strandings (n = 72) from 14–18 November. This date range corresponds with the dates the drifters shown in part (A) stranded. Red indicates the highest number of data points, while yellow indicates intermediate and blue indicates the lowest number.

Discussion

Expanding on previous research by Liu et al. (2019) and Santos et al. (2018), we incorporated sea turtle-shaped surface and bottom (drogued) drifters that were more representative in size and shape of individuals in the study population into our study of the currents in Cape Cod Bay. We found that the sea turtle-shaped surface drifter model behaved distinctly different from the Davis drifters that were previously used to study currents in Cape Cod Bay and sea turtle stranding locations (see Liu et al., 2019). However, as the distance between the drifters increased, so did the variability between the trajectories of the surface and bottom drifters. For example, if the surface drifter entered the longshore current while others were still in deeper water, we could no longer compare their paths directly since they were in very different regions and water masses. This is the reason we chose to limit our analysis to roughly the first 4 h after deployment.

It is also important to note that the different drifter design elements could have influenced the drift pattern of the sea turtle drifters. For instance, the hinged flippers on the sea turtle surface drifter could have added drag, while the choice of ballast weight—although the weight was comparable to the recorded weights of similarly sized cold-stunned sea turtles—could have exposed more of the carapace than what may be observed in a cold-stunned sea turtle. This difference in buoyancy would increase the influence of wind on the drifters and future research will correct the drifter design to account for this. The weight of the ballast on the bottom drifter also caused loss of data for some drifters because the buoys were pulled below the surface preventing communication with the satellite. Additionally, although the weight of the buoy attached to the bottom drifter was insignificant compared to the weight of the drifter, we cannot rule out the possibility that the surface area of the buoy added to the effect of the wind on the drifter.

Our analysis showed an overlap between the stranding locations of the sea turtles and drifters, although the proximity of the drifter deployment location to the shore likely added to the difference in stranding hotspots (i.e., drifters vs. turtles). We also noted that the stranding hotspots for the bottom drifters were south of the hotspots of the surface drifters. These results were consistent with what we generally know about variation in currents with depth in the Northern Hemisphere—because of friction and the Coriolis effect, deeper currents flow to the right of the wind direction in a process called an Ekman spiral.

The difference in stranding time between the surface and bottom drifters—bottom drifters taking approximately 10 times as long to strand—may provide clues as to the timing and condition (alive or deceased) of cold-stunned sea turtles. Cold-stunned sea turtles strand in pulses following cold fronts—with earlier waves arriving mostly alive and later mostly deceased—and these data could suggest that the turtles that strand later could have been submerged.

Another important observation was that two drifters, both deployed on 31 October—one during a pilot study in 2018 (see Supplemental Information) and one during this study in 2019—drifted out of Cape Cod Bay and into the Atlantic Ocean. On both occasions, they were deployed during the cold-stun stranding season, which begins mid-October, but before the first dramatic seasonal change in weather. This could indicate that, even if cold stunning occurs early in the season, some turtles may be pushed out into the open waters of the Atlantic Ocean rather than becoming trapped in the bay—likely increasing their chances of survival if they reach the warmer waters of the Atlantic.

Experiments of this sort in the future might include deployment locations throughout the bay. While we do not know where in the bay sea turtles lost mobility, there were several days (22%) when cold-stunned sea turtles were found near the Davis drifters and bottom and surface sea turtle drifters. When team members were sent to recover the satellite-tracked drifters, approximately seven cold-stunned sea turtles were found—three live turtles near a bottom drifter and three (two live and one dead) near a Davis drifter. Also, while searching for stranded turtles, rescue teams found four beached surface sea turtle drifters near approximately 15 live cold-stunned sea turtles on one day and three more sea turtle drifters near two live cold stunned turtles on another. One caveat, however, is that this may be entirely coincidental because the small sample size of our drifter deployments limits the conclusions we can make about the correlation between drifter stranding locations and sea turtle stranding hotspots.

As described by Liu et al. (2019), particle tracking can be conducted through numerical ocean models to estimate the origin of cold-stunned turtles. However, more experiments need to be conducted with particles in different layers of the water column. As shown in our study, water parcels, and therefore free-drifting turtles, will be transported to different regions of the coast depending on the depth of water at their point of origin.

Conclusion

Previous research on the relationship between drifter data and stranding locations addressed several knowledge gaps but did not wholly capture the conditions experienced by cold-stunned sea turtles. However, this study developed and tested drifter models—a sea turtle-shaped surface drifter similar to the one used by Santos et al. (2018) and a new sea turtle-shaped bottom drifter—that more closely simulate the movement of immobilized cold-stunned sea turtles in Cape Cod Bay. These findings serve to advance our understanding of sea turtle drift trajectories, particularly for the individuals that sink to the bottom upon stunning, a group that has received little attention. This new information may help to inform conservation efforts focused on the recovery of cold-stunned sea turtles in Cape Cod Bay.

The variability of the currents in Cape Cod Bay makes it inherently difficult to predict stranding locations for turtles not floating at the surface, but the information gathered by this study will help expand search efforts by demonstrating that stranding locations vary depending on whether sea turtles are floating at the surface or below after cold-stunning. Also, taking into consideration the differences we observed in stranding hotspots for drifters and sea turtles, further research is needed to compare stranding locations to different drifter deployment locations throughout the bay, ideally to simulate different cold stunning locations, including where turtles are located before they cold stun. Understanding the environmental correlates driving sea turtle strandings, both at the surface and bottom, will increase the likelihood of more quickly recovering cold-stunned sea turtles in Cape Cod Bay, thereby increasing the chances of survival.

While cold stunning is only one of the many threats to critically endangered Kemp’s ridley sea turtles, it is one of the most crucial threats to the thousands of juvenile sea turtles foraging in the Northwest Atlantic region. The information gathered by this research brings us closer to identifying the pathways of transport for cold-stunned turtles through both the surface and bottom currents—one puzzle piece at a time.

Supplemental Information

Supplemental Information 1 Comparison of sea turtle-shaped drifter and cold-stunned sea turtle stranding hotspots in Cape Cod Bay, Massachusetts from 19–26 November, 2019.

(a) Drifters (n = 6) deployed on 19 November. (b) Cold-stunned sea turtle strandings (n = 66) from 19–26 November. Red indicates the highest number of data points, while yellow indicates intermediate and green indicates the lowest number.

Click here for additional data file.

Supplemental Information 2 Comparison of sea turtle-shaped drifter and cold-stunned sea turtle stranding hotspots in Cape Cod Bay, Massachusetts from 26 November–02 December, 2019.

(a) Drifters (n = 5) deployed on 26 November. (b) Cold-stunned sea turtle strandings (n = 72) from 26 November–02 December. Red indicates the highest number of data points, while yellow indicates intermediate and green indicates the lowest number.

Click here for additional data file.

Supplemental Information 3 Data points for miniature sailboat deployed on 14 November 2019.

Click here for additional data file.

Supplemental Information 4 Data points for surface sea turtle drifter (ID# 21) deployed on 14 November 2019.

Click here for additional data file.

Supplemental Information 5 Data points for surface sea turtle drifter (ID# 22) deployed on 14 November 2019.

Click here for additional data file.

Supplemental Information 6 Data points for surface sea turtle drifter (ID# 23) deployed on 14 November 2019.

Click here for additional data file.

Supplemental Information 7 Data points for Davis-style surface drifter deployed on 14 November 2019.

Click here for additional data file.

Supplemental Information 8 Data points for subsurface sea turtle drifter deployed on 14 November 2019.

Click here for additional data file.

Supplemental Information 9 Data points for miniature sailboat deployed on 19 November 2019.

Click here for additional data file.

Supplemental Information 10 Data points for sea turtle surface drifter (ID# 41) deployed on 19 November 2019.

This is part one of two of the dataset for this drifter.

Click here for additional data file.

Supplemental Information 11 Data points for sea turtle surface drifter (ID# 41) deployed on 19 November 2019.

This is part two of two of the dataset for this drifter.

Click here for additional data file.

Supplemental Information 12 Data points for sea turtle surface drifter (ID# 42) deployed on 19 November 2019.

Click here for additional data file.

Supplemental Information 13 Data points for sea turtle surface drifter (ID# 43) deployed on 19 November 2019.

Click here for additional data file.

Supplemental Information 14 Data points for Davis-style surface drifter deployed on 19 November 2019.

Click here for additional data file.

Supplemental Information 15 Data points for subsurface sea turtle drifter deployed on 19 November 2019.

Click here for additional data file.

Supplemental Information 16 Data points for the miniature sailboat deployed on 26 November 2019.

Click here for additional data file.

Supplemental Information 17 Data points for sea turtle surface drifter (ID# 62) deployed on 26 November 2019.

Click here for additional data file.

Supplemental Information 18 Data points for Davis-style surface drifter deployed on 26 November 2019.

Click here for additional data file.

Supplemental Information 19 Data points for subsurface sea turtle drifter deployed on 26 November 2019.

Click here for additional data file.

Supplemental Information 20 2018 Davis drifter tracks.

These data show the hourly point locations for the Davis drifter that was deployed on 31 October 2018 before eventually being lost to sea. This is the unpublished data mentioned in line 324-325.

Click here for additional data file.

Supplemental Information 21 Weather data collected at Provincetown Municipal Airport, Provincetown, Massachusetts.

This data was used to estimate wind speed around the time of the drifter deployments.

Click here for additional data file.

Supplemental Information 22 Sea and weather data from NOAA National Data Buoy Center station 44018.

This buoy is located north of the drifter deployment site north of Provincetown, Massachusetts. The data was used to estimate wind speed around the time of the drifter deployments.

Click here for additional data file.

Supplemental Information 23 Sea and weather data from NOAA National Data Buoy Center station 44090.

This buoy is located west of the drifter deployment site in Cape Cod Bay, Massachusetts. The data was used to estimate wind speed around the time of the drifter deployments.

Click here for additional data file.

Supplemental Information 24 2019 sea turtle cold stunning data.

This dataset shows the locations and species ID of cold-stunned sea turtles were recovered during the 2019 stranding season.

Click here for additional data file.

Supplemental Information 25 Python algorithm for calculating the direction the drifters travelled.

Click here for additional data file.

Supplemental Information 26 Python algorithm used to calculate the distance between data points for the drifter datasets.

This information was used to estimate the drift speed.

Click here for additional data file.

We thank captain Chip Carroll and the crew of the F/V Albatross for assistance with drifter deployments; Ryan Page for educational outreach collaborations and help with drifter construction; students from regional schools and non-profit organizations working with at-risk youth for help building, decorating, and deploying drifters; and the Sea Turtle Stranding and Salvage Network volunteers who helped recover stranded drifters. We also thank our reviewers for the thoughtful and thorough feedback that helped improve this manuscript.

Additional Information and Declarations

Competing Interests

Author Contributions

Field Study Permissions

Data Availability

The authors declare that they have no competing interests.

Felicia M. Page conceived and designed the experiments, performed the experiments, analyzed the data, prepared figures and/or tables, authored or reviewed drafts of the article, and approved the final draft.

James Manning conceived and designed the experiments, performed the experiments, authored or reviewed drafts of the article, and approved the final draft.

Lesley Howard performed the experiments, analyzed the data, authored or reviewed drafts of the article, and approved the final draft.

Ryan Healey analyzed the data, authored or reviewed drafts of the article, and approved the final draft.

Nancy E. Karraker conceived and designed the experiments, authored or reviewed drafts of the article, and approved the final draft.

The following information was supplied relating to field study approvals (i.e., approving body and any reference numbers):

The U.S. Fish and Wildlife Service (issued to the Massachusetts Audubon Society) approved the study (Recovery Permit #1150C-1).

The following information was supplied regarding data availability:

Code and raw data are available in the Supplemental Files.

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
