# Peer review of "Developing bottom drifters to better understand the stranding locations of cold-stunned sea turtles in Cape Cod Bay, Massachusetts"

_PeerJ, doi:10.7717/peerj.15866_

## Round 0.1 · original submission · Major Revisions

Dear authors,

The reviewers have recommended revisions to your manuscript. Therefore, I invite you to respond to the reviewers' comments and revise your manuscript. Please take particular care of the comments of reviewer #3 provided an attached PDF file.

·

Basic reporting

I enjoyed reading this manuscript, it was clear, professional and well written.

It seems that the authors may not have been aware of some relevant literature references that focused on sea turtle drift research that they could have used to for comparison or in the Discussion. Cook et al. 2021 used wooden effigies as well as the same cold-stunned sea turtles that the authors are investigating.

Cook, M., J. Reneker, R. W. Nero, B. A. Stacy, D. S. Hanisko and Z. Wang. 2021. Use of drift studies to understand seasonal variability in sea turtle stranding patterns in Mississippi. Frontiers in Marine Science 8:659536. doi.org/10.3389/fmars.2021.659536

Reneker, J. L., M. Cook and R. W. Nero. 2018. Preparation of fresh dead sea turtle carcasses for at-sea drift experiments. NOAA Technical Memorandum NMFS-SEFSC-731. 14 pp. doi:10.25923/9hgx-fn38

These focus on the modeling aspect but could still provide useful information.
Nero, R. W., M. Cook, A. T. Coleman, M. Solangi, and R. Hardy. 2013. Using an ocean model to predict likely drift tracks of sea turtle carcasses in the north central Gulf of Mexico. Endangered Species Research 21:191-203. doi: 10.3354/esr00516

Nero, R. W., M. Cook, J. L. Reneker, Z. Wang, E. A. Schultz and B. Stacy. (2022). Decomposition of Kemp's ridley (Lepidochelys kempii) and green (Chelonia mydas) sea turtle carcasses and its application to backtrack modeling of beach strandings. Endangered Species Research. Endangered Species Research 47:29-47. doi.org/10.3354/esr01164

The article had professional article structure, included relevant figures, however, Figs 9 & 10 could probably be in the appendix or supplemental if space is an issue as they are similar to Fig. 8. The tables were appropriate and the raw data was shared. The article followed the required standard format.

Experimental design

The study expands on previous work conducted by Liu et al. 2019 to further investigate a very important issue that can affect sea turtle, specifically Kemp’s ridley, conservation. The ability to predict where cold-stunned sea turtles will beach to expedite response efforts would greatly increase the turtle’s chance for survival as well as make the best use of volunteer manpower. This is the first study that used submerged sea turtle drifters to investigate how those incapacitate animals may drift and eventually beach. The approach is novel and unique.
The Methods were described in detail, however, there was information missing on the type of GPS and Satellite transmitted which would be required for another investigator to replicate.

I have included additional comments within the manuscript.

Validity of the findings

I think this was a good, well planned research study. I suggested some additional references that I think the authors may have not been aware of and have included questions and comments in the manuscript. I would like to see the authors expand the discussion and conclusion think about how their conclusions could be applied to cold-stunned sea turtles. You discovered that the submerged drifters were 10x slower than the surface turtle drifters. What do you think that means if the same is true of live cold-stunned sea turtles? You also observed that turtles were lost to the Atlantic. That is another very important finding that could be discussed.

They only were able to use data from 3 of the 6 deployments but they did not mention why 3 were not useable. The most important comparisons are those of the sea turtle surface and subsurface drifters, could that data be used? I recommend the manuscript for publication with revisions that address my questions and expansion of the discussion/conclusion.
The authors need to add more to the discussion and

Additional comments

Additional comments are in an edited version of the manuscript pdf.

·

Basic reporting

No comment.

Experimental design

No comment.

Validity of the findings

No comment.

Additional comments

This is a useful study providing valuable information on how debilitated sea turtles are likely to move with water currents and winds and thus help predict where they are likely to wash ashore. Such information is very useful to help organize search/rescue operations for cold-stunned sea turtles.

In the Introduction, it may be worth further contextualizing this work by noting that the conservation efforts that have led to sea turtle population increases will necessarily mean that a greater number of turtles will be exposed to threats (compared to historical data - strandings will increase, bycatch will increase, etc.). This is not necessarily cause for alarm, but it does require further considerations as to how sea turtles are effectively managed. See, for instance, the article “Managing fisheries in a world with more sea turtles” https://royalsocietypublishing.org/doi/10.1098/rspb.2020.0220

One of the most important points to more clearly make in this paper is that, given the unknown initial distribution of turtles and the non-random/relatively small sample size of drifter deployments, there is no reason to expect a close correspondence between turtle stranding hotspots and drifter stranding locations.

It is probably also worth mentioning similar work related to that by Nero and Cook in the northern Gulf of Mexico on Kemp’s ridley strandings:

Nero, R.W., Cook, M., Coleman, A.T., Solangi, M. and Hardy, R., 2013. Using an ocean model to predict likely drift tracks of sea turtle carcasses in the north central Gulf of Mexico. Endangered Species Research, 21(3), pp.191-203.
https://www.int-res.com/abstracts/esr/v21/n3/p191-203/

Nero, R.W., Cook, M., Reneker, J.L., Wang, Z., Schultz, E.A. and Stacy, B.A., 2022. Decomposition of Kemp’s ridley (Lepidochelys kempii) and green (Chelonia mydas) sea turtle carcasses and its application to backtrack modeling of beach strandings. Endangered Species Research, 47, pp.29-47.
https://www.int-res.com/abstracts/esr/v47/p29-47/

Cook, M., Reneker, J.L., Nero, R.W., Stacy, B.A., Hanisko, D.S. and Wang, Z., 2021. Use of drift studies to understand seasonal variability in sea turtle stranding patterns in Mississippi. Frontiers in Marine Science, 8, p.447.
https://www.frontiersin.org/articles/10.3389/fmars.2021.659536/full

Reviewer 3 ·

Basic reporting

All comments are within a single PDF submitted here,

Experimental design

See attached

Validity of the findings

see attached

Annotated reviews are not available for download in order to protect the identity of reviewers who chose to remain anonymous.

---

## Round 0.2 · Minor Revisions

Dear Authors,

There are still some minor comments to be addressed, and once you address them I should be able to accept the revision without further reviewer input.

Best wishes,

·

Basic reporting

It is obvious that the authors read, considered and incorporated the comments from the previous 3 reviewers. The manuscript is improved and I enjoyed reading it. Unfortunately, for the authors they were stuck with a reviewer that has done very similar research so I did come up with a few more suggestions and questions. But nothing that should take too long to address.

I should have asked this the first time, do you think the subsurface turtle drifter was basically drifting along the bottom or was it in the water column? You stated you picked a location where the drifter could reach the bottom of the bay (line 210). I am just wondering if they should be called bottom drifters to better describe what is really happening. Sub-surface, to me, has it drifting in the water column within a few meters of the surface. The drifter would be neutrally buoyant if it were sub-surface and these were negatively buoyant (line 188). The Davis drifter is more of a subsurface drifter.

Experimental design

I realized that there is no statistical analyses in this manuscript when there could be. At a minimum they could compare the drift time speed (table 1 data) to see if there was a significant difference between each object.

Validity of the findings

I've added a few more comments/questions that once considered and answered would enhance the findings of this study.

Please double check figure 5. I think the surface drifter data is missing.

Additional comments

Once again, I enjoyed reading this manuscript and I hope you find that my additional comments and suggests were helpful and only improve the great work you have done.

Reviewer 3 ·

Basic reporting

Since I've already done all this once, I'm not taking the time to do it again in detail. This is a significant improvement over the initial submission. See the PDF attachment for short set of new comments.

Experimental design

no comment

Validity of the findings

no comment

Additional comments

no comment

Annotated reviews are not available for download in order to protect the identity of reviewers who chose to remain anonymous.

---

## Round 0.3 · accepted · Accept

Dear Dr. Page,
Thank you for your submission to PeerJ.

I am writing to inform you that your manuscript - Developing bottom drifters to better understand the stranding locations of cold-stunned sea turtles in Cape Cod Bay, Massachusetts - has been accepted for publication.